# From Interpolation to Extrapolation: Complete Length Generalization for Arithmetic Transformers

## Abstract

Since its introduction, the transformer model has demonstrated outstanding performance across various tasks. However, there are still unresolved issues regarding length generalization, particularly in algorithmic tasks. In this paper, we investigate the inherent capabilities of transformer models in learning arithmetic algorithms, such as addition and multiplication. Through experiments and attention analysis, we identify a number of crucial factors for achieving optimal length generalization. We show that transformer models are able to generalize to long lengths with the help of targeted attention biasing. We then introduce Attention Bias Calibration (ABC), a calibration stage that enables the model to automatically learn the proper attention biases, which we link to mechanisms in relative position encoding. We demonstrate that using ABC, the transformer model can achieve unprecedented *perfect* length generalization on certain arithmetic tasks. [1]

## 1 Introduction

The Transformer architecture has been the fundamental building block of many SOTA solutions across a wide range of machine learning tasks, including recent phenomenal Large Language Models (LLMs). However, one serious issue with today's Transformer models is length generalization, or *extrapolation*, defined as "a model's ability to continue performing well as the number of input tokens during validation increases beyond the number of tokens on which the model was trained" (Press et al., 2022). Many models achieve good accuracy with small inputs but fail to produce meaningful results with long inputs (Ruoss et al., 2023). The problem has been extensively studied in language models, as extending the context window sizes of LLMs is an urgent and hot topic (Anil et al. (2022); Chi et al. (2023); Chen et al. (2023)).

In this paper, we investigate Transformer's performance in learning arithmetic algorithms, which presents a unique context for studying length generalization. Deletang et al. (2023) relate sequence prediction to formal language theory by treating sequences as elements of infinite formal languages generated by a grammar. Many sequence-to-sequence tasks can be seen as language transduction tasks, where the goal is to learn the underlying generation rules. Arithmetic tasks fall in this paradigm and the existence of such rules provides a convenient setting where we can examine the internal mechanisms of the model.

Thus our goal is to obtain complete generalization and study the enabling factors during the process. There are specially constructed architectures such as Chiang & Cholak (2022); Deshpande et al. (2021) that achieve generalization on some of the tasks we study. However, like Deletang et al. (2023), our focus is on the conventional Transformer architecture and regular training process (e.g., SGD and variants), since this setting is the case for many SOTA models, especially those successful LLMs. Along this line, results obtained by previous works are rather gloomy. According to the extensive study (20910 models, 15 tasks) conducted by very recent work Deletang et al. (2023), the Transformer can only solve one Regular and one CS task in the Chomsky hierarchy, and fail all others, regardless of the positional encoding. [2] Similar results are obtained by Ruoss et al. (2023). This is echoed by the theoretical difficulties discovered in Hahn (2020).

---

[1] We will open source our code upon the publication of this paper.
[2] And their standard of "solving" a task is above 90% accuracy averaged across all lengths.

Length generalization for arithmetic tasks is essentially inductive inference since the ultimate goal is learning a general rule from a finite number of observations. It is known that inductive inference requires inductive biases (Mitchell, 1980), additional assumptions that are independent of the data. This is because any finite number of training samples has infinite possible continuations corresponding to different generation rules.

In this work, we draw attention to the stage of model *interpolation*. Interpolation is a special form of in-distribution generalization, and we define it as a model's ability to perform well on examples that are novel but with lengths within the same range as those samples from the training set. We show in section 5 that the patterns that the model acquires during the interpolation stage can be used as a form of inductive biases to re-train the model to achieve *extrapolation*.

The contributions of this work include:

- We are the *first* Transformer-based architecture to obtain complete generalization on a number of arithmetic tasks: successor function, parity, addition, and a restricted version of multiplication. Our models produce results with 100% accuracy up to 50 digits.
- We show that, for the tasks that we study, (the right) attention is indeed all you need. Transformer can perform well as long as it attends to the right tokens. And we identify a few key factors in achieving such proper attention.
- Based on our findings, we introduce *attention bias calibration* (ABC), a process that automatically collects attention patterns learned from training data and extends them to long lengths. We show that this automizes the above mechanisms. In addition to that, we show ABC's relation to RPE, which opens the potential for its applications to more complicated tasks.

Figure 1 summarizes the generalization that our ABC scheme achieves on two of the tasks, with comparisons against popular alternatives such as ALiBi, RoPE, etc. ABC is the only solution achieving perfect generalization. We obtain similar results on other tasks which will be discussed in detail in section 6.

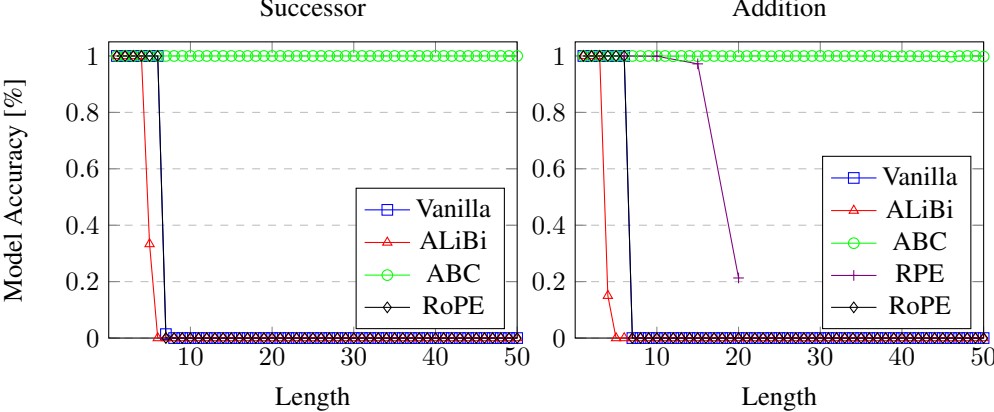

Figure 1: Extrapolation results for models trained on $L_{int} \leq 6$ on **Successor** and **Addition**. Length is measured in the number of digits of one operand.

## 2 RELATED WORK

Besides the line of work on arithmetic algorithms learning mentioned earlier, length generalization for Transformers is a also very hot topic in other areas. Existing approaches have been focusing on two aspects: positional encoding (PE) or/and attention bias (AB).

**Relative Position Encoding**. Relative position encoding (RPE) relies on the relative distance between tokens to construct position embeddings. This approach is first proposed by Shaw et al. (2018) and has shown to produce significant improvements over absolute positional encoding in machine

translation tasks (Shaw et al., 2018). This leads to its application in numerous machine learning models and the development of multiple variations such as Transformer-XL (Dai et al., 2019) and RoPE (Su et al., 2022).

**Attention Biasing**. Attention biasing, on the other hand, adds a bias directly to the attention matrix, allowing the model to extrapolate to longer lengths efficiently. First introduced as ALiBi (Attention with Linear Biases) by Press et al. (2022), it is quickly followed by similar models such as KER-PLE (Chi et al., 2022), and Sandwich (Chi et al., 2023), all showing certain improvement in length extrapolation. Other forms of biases include sliding window (Beltagy et al., 2020) and its variations. Compared to other relative position encoding schemes, attention biasing typically demands less computational resources.

These two lines of work are closely related and there are extensive studies on their effectiveness. [3] However, the results are mixed. On one hand, the popular belief is that relative PEs (Shaw et al., 2018; Dai et al., 2019; Su et al., 2022) are more effective in length generalization than absolute variants (Vaswani et al., 2017). On the other hand, however, some works (e.g., Kazemnejad et al. (2023)) point out that such a conclusion is obtained by using language modeling perplexity as the sole metric, which may not reflect actual performances on downstream tasks. In fact, Kazemnejad et al. (2023) show that, on a collection of reasoning and mathematical tasks, No Positional Encoding (NoPE) actually performs the best. Likewise, Deletang et al. (2023) show that state-of-the-art PE or AB methods do not help Transformer extrapolate on arithmetic tasks.

Our work is also related to the line of research studying the systematic generalization capabilities of Transformers (Ontanon et al., 2022) which show that various techniques such as embedding scaling or early stopping (Csordás et al., 2021), copy gate and geometric attention (Csordás et al., 2022), and randomized positional encoding (Ruoss et al., 2023) are all effective.

## 3 EXPERIMENT SETUP

### 3.1 TASKS

Let $\mathbb{N} = \{0, 1, 2, \ldots\}$ be the set of natural numbers. We consider the following arithmetic tasks:

- **Successor function**: Maps a natural number to the next one: $S(n) = n + 1$ for $n \in \mathbb{N}$.

- **Addition**: $y = x_1 + x_2$ for $x_1, x_2 \in \mathbb{N}$.

- **Parity**: Given $x \in \mathbb{N}$, this operation returns 1 if the binary representation of $x$ contains an odd number of 1's, and 0 otherwise.

- $N \times 1$: $y = x_1 \times x_2$ for $x_1 \in \mathbb{N}$ and $x_2 \in \{0, 1, \ldots, 9\}$. This is a restricted form of multiplication where one of the operands is restricted to single-digit.

These tasks are well-known examples in the theory of computation. The seemingly trivial `Successor` function is the basic component of Peano axioms, which formalize the structure of the natural numbers. Using the digit-based representation, `Successor`, `Addition` and $N \times 1$ all belong to the Type-1 context-sensitive (CS) category of the Chomsky hierarchy. `Parity`, on the other hand, is in Type-3 Regular (R) category (Deletang et al., 2023), since it can be solved with a 2-state finite-state machine. $N \times 1$ is a task that is specifically constructed to test if the methods we develop could be extended to more complex operations such as multiplication. Restricting one operand to be single-digit results in a simpler attention pattern that allows for easy analysis.

We also conduct experiments on more complex tasks such as multi-digit multiplication and ListOps (Nangia & Bowman, 2018). The latter is a kind of prefix arithmetic that tests a model's tree learning capabilities. It turns out that Transformers with the scale that we study are unable to learn such tasks. They serve as indications of the limitations of our current solution and we discuss their implications in section 8.

---

[3]Please see Dufter et al. (2022) for a comprehensive review of methods to incorporate position information into Transformer models.

### 3.2 TOKENIZATION AND PROBLEM REPRESENTATION

Unlike some previous works that study the arithmetic operations within a finite group and treat each number as a token (e.g., Power et al. (2022)), we use a character-based tokenizer which allows us to represent an infinite number of integers using a finite set of tokens, enabling unbounded extrapolation testing. Thus in our study the numbers and operators are all encoded in their natural form and we use a vocabulary of 15 tokens: {0, ..., 9, +, *, $, &, @}, where the last three characters represent SOS, EOS, and PAD, respectively.

With our tokenization, all tasks are naturally sequence-to-sequence except for **Parity** which is classification. We turn **Parity** into a sequence-to-sequence task as follows. Let $x_n \ldots x_1$ be the input sequence of a binary string where $x_i \in \{0, 1\}$, the target is generated as $y_1 = x_1, y_i = y_{i-1} \otimes x_i$ for $i = 2, \ldots, n$, where $\otimes$ denote bitwise xor.

To facilitate learning, we pad each operand with 0's to the left to a fixed length. In addition to that, we reverse the ordering of the tokens in the output sequence to match the natural generation process. For example, $0123 + 0748 \rightarrow 1780$.

### 3.3 MODEL CONFIGURATION

Since our tasks are sequence-to-sequence, we choose an encoder-decoder architecture, with 1 encoder layer and 6 decoder layers, all with 8 attention heads. The embedding size is 128 and feed forward size 512. We tried models with a number of different sizes and found no significant difference across all variations that could converge. We settled on the model above and did not pursue the configuration with the optimal size.

We train our models using cross-entropy loss and Adam optimizer, with learning rate $10^{-5}$ and a dropout of 0.3. For training for interpolation, we generate a random permutation $\Pi$ of numbers in the range $[0, 2^{20}]$ and split the set by a 7:1 ratio for training and validation. For binary operations such as **Addition**, both operands are drawn independently from $\Pi$. Thus both the training and validation data sets consist of mainly 6-digit numbers, in base 10, with less than 5% 7-digit numbers. We denote $L_{int}$ the length of input, measured as the maximum number of digits in the operand(s), during the interpolation phase. Note that length refers to the number of digits in the operands, not the total input sequence length.

We define successful extrapolation as the model's ability to achieve near-perfect accuracy on inputs with lengths up to 50 digits. For extrapolation testing, for each length $L$, we randomly sample $\min(base^L - base^{L-1}, 10000)$ numbers of length $L$ and compute the accuracy on these samples, where $base$ is 10 for decimal tasks [4] and 2 for binary ones. The model's output is considered accurate if and only if it exactly matches the correct label sequence.

We use greedy decoding for all inferences.

## 4 (THE RIGHT) ATTENTION IS ALL YOU NEED

To develop our ideas, we first train vanilla Transformers with some commonly used length generalization methods, including the original sinusoidal positional encoding, ALiBi, and RoPE. The results on **Successor** and **Addition**, together with the performance of our ABC scheme, have been shown in figure 1 at the beginning of the paper. All models achieve some levels of interpolation but none could extrapolate beyond training length. RoPE and vanilla Transformer perform almost identically, dropping precipitously to almost 0 accuracy once the length goes beyond 6. We observe similar patterns with other tasks.

To figure out the causes of failure to extrapolate, we extract and analyze the attention weights of the vanilla model on **Successor** and **Addition**. Figure 2 shows the attention heat maps of one specific head in the last decoder layer when decoding **Successor** and **Addition** tasks. Lighter colors represent higher weights. More detailed analysis is presented in appendix A.3 but the patterns are very clear: the vanilla Transformer correctly learns the right attention patterns up to the training

---

[4]We actually experimented with different bases, such as 4, 8, 16, etc., for those non-binary tasks. Our code and the techniques all work seamlessly. For simplicity we only report results on base 10 cases in this paper.

length and fails beyond that. This correlates perfectly with the extrapolation performance shown in figure 1.

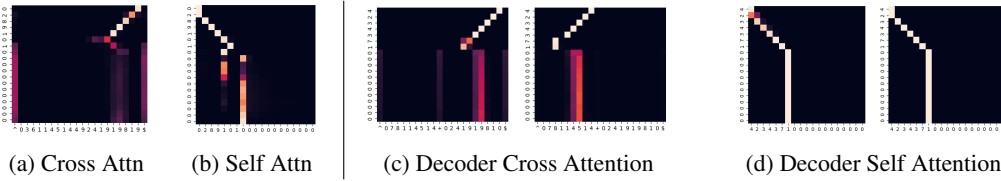

| (a) Cross Attn | (b) Self Attn | (c) Decoder Cross Attention | (d) Decoder Self Attention |

Figure 2: Attention heat maps for **Successor** (Left) and **Addition** (Right).

## 4.1 ATTENTION BIAS SCAFFOLDING

In this section, we introduce a number of methods that guide the model to attend to the right places. Assisting model learning is a common practice. Relevant techniques include inputs combination (Libovický et al., 2018), "arithmetic prompting" (Zhou et al., 2022), representation transformation (Nogueira et al., 2021), scratch pad (Lee et al., 2023), etc. Indeed, most of our methods are drawn from these toolboxes as well. However, we use them to target directly at the attention, thus we call our approach Attention Bias Scaffolding (ABS). In the following we briefly summarize the two most effective ones. A detailed treatment including visualizations can be found in appendix A.4.

**Windowed Attention Biasing**. The idea was developed by Lonformer (Beltagy et al., 2020). The basic intuition is that the most important local dependency is typically restricted by a limited range. In the context of arithmetic algorithm learning, often local context is the *sole* determining factor, [5] which can be captured by a sliding window of width $w$ (Beltagy et al., 2020).

**Cyclic Position Indexing (CPI)**. Position indexing refers to how we identify each individual position. The simplest way is just to index them $0, 1, \ldots$. As our tasks have very restricted dependency contexts which are localized by the windowed attention biasing, the model only needs a way to differentiate positions within the context window thus long position indexing is not necessary. And our empirical study shows that it can be harmful sometimes. Therefore we propose Cyclic Position Indexing (CPI): Let $i$ be the position index of a token. We select a period parameter $T$ and convert token positions to $i \mod T$ before entering into the model. Essentially indices are cycling around back when they get large.

## 4.2 RESULTS OF ABS

To evaluate the effectiveness of the above mechanisms, we conduct extensive experiments on each of the arithmetic tasks with a number of their combinations. Results and detailed discussion are presented in table 3 in appendix A.5. We summarize the main findings here:

- None of the previous works achieves extrapolation on any of the tasks.
- Our solutions (windowed attention biasing + CPI) achieve complete length generalization on all tasks, maintaining 100% accuracy up to 50 digits.
- Unary tasks (**Successor** and **Parity**) appear to be not relying on any positional embedding at all once the windowed attention biasing is in place.
- For binary tasks (**Addition** and $N \times 1$), there appears to be some bad interaction between the original sinusoidal PE and windowed attention biasing. Their combination achieves only interpolation but not extrapolation.

**The Case of Parity**. **Parity** is a well-known failure mode for Transformer due to the theoretical limitation found by Hahn (2020). Deletang et al. (2023) prove that there exists a Transformer construction that can achieve perfect Parity, but such a construction is not learnable. Ours is the *first* learnable Transformer that obtains perfect accuracy and length generalization for the **Parity** task, overcoming the difficulties of Hahn (2020) via attention scaffolding.

---

[5]This is why the arithmetic calculation algorithms that we humans use could be applied to arbitrarily long inputs.

The above analysis and experiments lead us to believe that the right attention is the key to achieving good generalization (thus the title of this section). PEs are just means to achieve proper attention. Methods that target directly at attention might produce superior results on some tasks.

## 5 ATTENTION BIAS CALIBRATION (ABC)

Having demonstrated the important role of correct attention in the transformer model's learning, we introduce Attention Bias Calibration (ABC), an automatic process that extends the working attention patterns of a model that achieves successful interpolation to arbitrary lengths while preserving its near-perfect performance. The idea is, a model trained to full interpolation must be able to produce the right attention pattern on interpolation data (see section A.3), which captures the local dependencies for recurrent arithmetic algorithms. ABC extracts and aggregates the attention weights and uses them as attention bias, like Press et al. (2022), to fine-tune the model for long inputs. Similar to the scaffolding in section 4.1, ABC is also a kind of inductive bias, but it is fully automatic.

Let $m \times n$ be the dimensions of the attention matrix of a model that has interpolated and $M \times N$ the dimensions that we would like to extrapolate to. It should hold that $m < M$ and $n < N$. ABC proceeds in the following steps:

**1. Training for Interpolation:** First we train a vanilla transformer model $\boldsymbol{T}_{int}$ on the dataset $\mathbb{S}_{int}$ until it is capable of interpolation. By this point, the accuracy of $\boldsymbol{T}_{int}$ should be near perfect. Then we use $\boldsymbol{T}_{int}$ to decode a random subset of training samples $\mathbb{S}_{gen} \subset_R \mathbb{S}_{int}$ and extract the attention weights. Because this process is identical for all heads, to simplify notation, we omit their indices. Let $x_k[i]$ be the embedding vector for the $i$-th token in sample $k$, the attention matrix is extracted as

$$A_{i,j}^k = x_k[i] \boldsymbol{W}^Q \boldsymbol{W}^{K^\top} x_k[j]^\top$$

where $\boldsymbol{W}^Q, \boldsymbol{W}^K$ are parameter matrices in the last decoder layer of model $\boldsymbol{T}_{int}$.

**2. Attention Biases Computation:** We then average the attention weights for all data in $\mathbb{S}_{gen}$:

$$\bar{\boldsymbol{A}} = \frac{1}{|\mathbb{S}_{gen}|} \sum_{k=1}^{|\mathbb{S}_{gen}|} \boldsymbol{A}^k$$

The next steps average attention weights along a number of lines within the elements of the matrix and extend them along those particular directions. We observe that attention patterns manifest themselves along lines of the attention matrix and these are the directions we expend them. Theoretically, we could explore any direction but empirically we find it suffices to try the diagonal, the anti-diagonal, and vertical lines. The figure below visualizes the said directions, with line sums annotated on the sides:

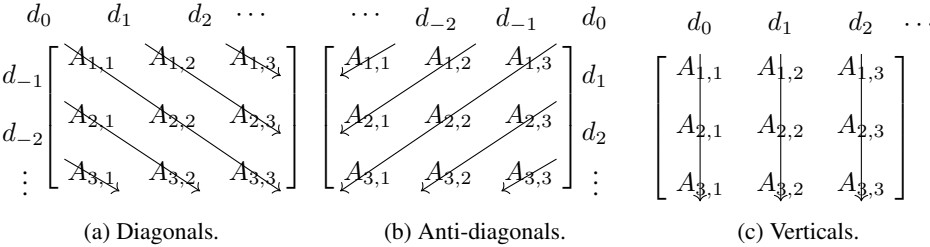

(a) Diagonals.      (b) Anti-diagonals.      (c) Verticals.

Figure 3: Examples of the different diagonals ABC can take

For all directions we consider, let $l$ be the set of elements on a line, we perform the following steps:

**2.1. Averaging across Lines:**

$$d_l = \frac{1}{|l|} \sum_{(i,j) \in l} \bar{A}_{i,j}$$

This step effectively "summarizes" each line into a single value.

**2.2. Bias Matrix Extension:** Next we extend $\bar{A}$ into any arbitrary size $\tilde{A} \in \mathbb{R}^{M \times N}$ via:

$$\tilde{A}_{i,j} = \begin{cases} dropoff(d_l - d_{max}), & \text{if } l \text{ exists in } \bar{A} \\ -\inf, & \text{otherwise} \end{cases} \tag{1}$$

where $d_{max}$ is the maximum value of $d_l$'s among all the lines of $\bar{A}$, and

$$dropoff(x) = \begin{cases} x, & \text{if } x > threshold \\ -\inf, & \text{otherwise} \end{cases}$$

What this process does is actually very simple: for the elements along the extension of existing lines of $\bar{A}$, it first subtracts $d_{max}$ from $d_l$, then cuts off at a $threshold$. Elements not on the extensions of $\bar{A}$'s lines will be set to $-\inf$. For our task, the dropout threshold is set to $2\sigma + \mu$, where $\sigma$ and $\mu$ are the standard deviation and the mean of all the $d_l$'s, respectively. This is a very strict threshold, meaning that it only preserves really strong patterns. For other tasks where patterns are not that obvious, a softer threshold value or even no dropout may be used.

**2.4. Finalization:** The final bias matrix $\tilde{A}$ is obtained by performing an element-wise $\max$ operation among the matrices from equation 1 across all directions. We then repeat for each of the heads, equipping them with independent biases. If the final bias matrix consists of only $-\inf$'s, meaning that no pattern is picked up, we replace every $-\inf$ with 0, effectively leaving it "transparent".

The complete and detailed algorithm is presented in appendix B.

**3. Re-training with Attention Biases:** After the attention biases for each head have been constructed, we train another model on the same input sequences $\mathbb{S}_{int}$ with the constructed attention biases added to the attention weights.

$$A_{i,j} = x_i \boldsymbol{W}^Q \boldsymbol{W}^{K\top} x_j^\top + \tilde{A}_{i,j}, \quad \tilde{\boldsymbol{A}} \in \mathbb{R}^{M \times N}$$

Note that in this work the bias matrices are obtained from the last decoder layer and applied to all layers during re-train. More flexible configuration such as per-layer bias could work better for more complex tasks.

## 6 MAIN RESULTS

A prerequisite of ABC is that the vanilla Transformer must be able to train to interpolate. Among the tasks we study, as discussed in section A.6, `Parity` is apparently a failure. Thus we implement the vanilla Transformer (with sinusoidal PE), ALiBi, RoPE, and ABC, and test on the rest of the tasks.

The accuracy vs. input length curves of different models on `Successor` and `Addition` have been plotted in figure 1 at the beginning of this paper. The overall performance on all tasks is summarized in table 1. We observe that ABC performs vastly superior to other models across all tasks, achieving near-perfect accuracies up to 50 digits.

Figure 4 and 5 visualize the cross attention bias matrices, one for each head, learned by ABC, for `Addition` and $N \times 1$, respectively. Since the most meaningful attention activities happen in cross-attention, where the model is attending to the input sequence, we do not show self-attention biases. Each color map is plotted using a colorbar scaling of $[\min_h(\tilde{A}_h), \max_h(\tilde{A}_h)]$ for each individual head. Head bias with a small variance will result in a "transparent" bias matrix with all 0's after drop-off, in which case the 0's are painted black. Note that addition is a binary operation so the input length is twice the output sequence thus the matrices in figure 4 are rectangles instead of squares.

---

[6]An encoder-only architecture with shared layers.

Table 1: Extrapolation results measured as percent accuracy (%). Numbers in bold show the best accuracies achieved for the corresponding input length limit.

| Task | Model | Length (Number of Digits) | | | | |
|---|---|---|---|---|---|---|
| | | 6 | 10 | 15 | 20 | 50 |
| Successor | Vanilla | **100.0** | 0.0 | 0.0 | 0.0 | 0.0 |
| | ALiBi | 1.3 | 0.0 | 0.0 | 0.0 | 0.0 |
| | RoPE | **100.0** | 0.0 | 0.0 | 0.0 | 0.0 |
| | ABC | **100.0** | **100.0** | **100.0** | **100.0** | **100.0** |
| Addition | Vanilla | **100.0** | 0.0 | 0.0 | 0.0 | 0.0 |
| | ALiBi | 0.0 | 0.0 | 0.0 | 0.0 | 0.0 |
| | RoPE | **100.0** | 0.0 | 0.0 | 0.0 | 0.0 |
| | RPE* | **100.0** | 99.9 | 97.2 | 21.3 | N/A |
| | ABC | **100.0** | **100.0** | **99.9** | **99.9** | **99.8** |
| $N \times 1$ | Vanilla | **100.0** | 0.0 | 0.0 | 0.0 | 0.0 |
| | ABC | **100.0** | **100.0** | **100.0** | **100.0** | **100.0** |

\* Data taken from Jelassi et al. (2023). [6]



Figure 4: ABC cross attention bias for `Addition`

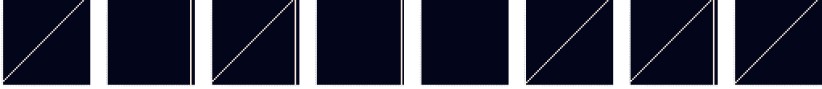

Figure 5: ABC cross attention bias for $N \times 1$

A few interesting patterns emerge. First, since the model generates output tokens in a reversed order, most of the open elements are along the anti-diagonal direction for both tasks. Second, there is a clear division of labor among the heads, which is consistent with the findings in A.3. More specifically, in `Addition`, heads 1, 4, 7 attend to the first operand, while the remaining heads attend to the second. In $N \times 1$, most heads attend to the multi-digit number and the multiplication sign while one of the heads, head 4, attends to the single-digit operand. Note that there are vertical lines in heads 1, 3, and 7 as well. Third, the different patterns show that our bias generation process is effective: the anti-diagonal and vertical patterns are learned by searching the $(1, 0)$ and $(1, -1)$ directions respectively, Note that there is an empty bias consisting of all 0s in 5 (head 5). This indicates that ABC did not pick up any patterns in that head.

**Running Time**. ABC requires a retraining stage. However, with the help of attention bias masks, this stage converges very fast. We observe that the time needed to retrain the model is only 1/100 to 1/10 of the time for training the model to interpolate.

## 7    ABC AS A GENERALIZED RPE

It turns out that ABC has a close tie to the relative position encoding (RPE) of Shaw et al. (2018) which has been shown to be a very robust PE and the foundation of many other variants (Dufter et al., 2022). Shaw et al. (2018) biases the attention at two places: (1) when computing the dot-product between query and key; and (2) when producing the weighted sum of value vectors. (2) has been shown to be not very useful (Shaw et al., 2018). Let $x_i$ be the embedding vector of the $i$-th

token, (1) is implemented as follows:

$$e_{ij} = \frac{(x_i \boldsymbol{W}^Q)(x_j \boldsymbol{W}^K + a_{ij}^K)^\top}{\sqrt{d_k}}$$

$$a_{ij}^K = w_{clip(j-i,k)}.$$

where $w$'s are a set of learned vectors and the bias vector $a_{ij}^K$ is selected from the set by a clipped indexing scheme: $clip(x, k) = \max(-k, \min(k, x))$. That is, tokens more than $k$ units from the current query token will be clipped to $k$. Note that the selection of $w$ vector depends solely on the relative distance between the query token $i$ and the key token $j$.

It is clear that both RPE and ABC bias the attention matrix. In the case of RPE, this is done by a vector inside the dot-product, whereas ABC achieves this with a scalar bias on the exterior. If we view elements in the bias matrices and which parameter determines each of them, then we can see the remarkable similarities between RPE and ABC. Figure 6 shows a comparison between attention bias matrices of RPE and ABC for the case extending along the diagonal. ABC averages along each of the $k$-diagonals at step 2.1 during its procedure. Thus for query $i$ and key $j$, the bias is $d_{j-i}$. The indexing scheme is exactly the same as that of RPE. And there is an implicit clipping too: for an attention matrix of dimensions $m \times n$ with $m \le n$, the set of possible $k$ values for valid $k$-diagonals are $\{-(m-1), -(m-2), \ldots, -1, 0, 1, \ldots, (n-1)\}$, a total of $m + n - 1$. When extending to $M \times N$, any elements outside those lines are set to $-\inf$. Effectively, this is an asymmetric clipping function: $clip(j - i, m - 1, n - 1)$.

$$\begin{bmatrix} w_0 & w_1 & w_2 \\ w_{-1} & w_0 & w_1 \\ w_{-2} & w_{-1} & w_0 \end{bmatrix} \qquad \begin{bmatrix} d_0 & d_1 & d_2 \\ d_{-1} & d_0 & d_1 \\ d_{-2} & d_{-1} & d_0 \end{bmatrix}$$

Figure 6: Factors determining bias weights in RPE (left) and ABC (right).

One difference between RPE and ABC is that RPE learns these parameters during training, whereas in ABC, the biases are calculated from correct interpolation results. In a sense, ABC is more general and, by scanning more directions, it has the potential to discover more regularities automatically.

## 8 DISCUSSION

It is important to clarify the scope and limitations of our work. Ours is an initial attempt to study the roles of attention in transformer's learning. And our definition of learning is achieving complete length generalization. To this end, we need to find tasks on which transformers can generalize. Surprisingly, even for tasks as simple as addition, no previous transformer model ever succeeded. We take this as an indication that our understanding of the model's learning mechanism is inadequate. Therefore, the successful cases obtained in this paper, either via ABS or ABC, even though only on simple recurrent patterns and task-specific models, solve a few long-standing difficult or even "impossible" tasks (e.g., Parity) and represent a significant step forward.

In order for ABC to work, the vanilla model must be able to interpolate, achieving near perfect accuracy within the training lengths. With the model scales we experimented and the amount of resources we had, we only made a few tasks work. Besides the tasks we have presented, we could only solve a simplified single-operator, single-depth ListOps task via ABS. Detailed solution is presented in appendix C. More complex tasks, such as multi-digit multiplication and multi-depth ListOps, appear to require composition of those simple patterns. Learning such compositions may require scaling up the model dramatically, as the heuristic in the NDR paper (Csordás et al., 2022) indicates. We leave them as future research.

There are a lot of opportunities for future exploration. The connection between ABC and RPE indicates that ABC could have potential applications in other fields such as NLP as well. We believe its potential in these fields is a direction worth investigating.

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

## A ATTENTION BIAS SCAFFOLDING DETAILS

In this section we provide more details on the importance of attention and our attention bias scaffolding methods. We develop our ideas through a series of initial experimentations, attention weights analysis, and final verification.

Existing approaches to optimizing length generalization of Transformer models have been focusing on two aspects: positional encoding (PE) or/and attention bias (AB). The two concepts are closely related. In fact, we believe they should be treated as two sides of the same coin: All PEs influence the attention, and almost all ABs, with the exception of no PE at all such as Kazemnejad et al. (2023) and ours, rely on position information to determine the bias. However, the best-performing AB methods' dependency on positional information is indirect: the bias is often determined by the distance between tokens, instead of their positions. Examples include ALiBi (Press et al., 2022) and RPE (Shaw et al., 2018). In addition, as our ABS and ABC schemes show, AB can work well without any position information. This is consistent with the findings of some previous works. For example, although Transformer's attention mechanism is order-invariant, decoder-only Transformers with causal attention mask are *not* and can model sequences without explicit position information (Tsai et al., 2019).

### A.1 OUR THOUGHTS AND FINDINGS

We have an interesting finding in a similar tune. That is, with our mechanism that enables the model to attend to the correct tokens, explicit position encoding is indeed not always necessary, even for achieving perfect generalization. With our architecture, cross-attention allows the model to attend to the correct input while self-attention relays the information from the previous step.

This leads us to believe that positional encoding or embedding is not the key to achieving good generalization. The right attention is. PE and AB are just means to attain the latter. Since there is no universal PE or AB that generalizes well on all tasks, for the tasks that we study in this work, auxiliary means that target directly at the attention could be used to achieve better generalization.

### A.2 INITIAL EXPERIMENTATION

To develop our ideas, we first train vanilla Transformers with some commonly used length generalization methods, including the original sinusoidal positional encoding, ALiBi, and RoPE, and examine the results.

Figure 7 shows the results on **Successor** and **Addition**. All models achieve some levels of interpolation but none could extrapolate beyond training length. Among them, RoPE and vanilla Transformer perform almost identically, dropping precipitously to almost 0 accuracy once the length goes beyond 6. Note that the RoPE implementation for **Addition** must use an embedding size of 512 otherwise it converges very slowly.

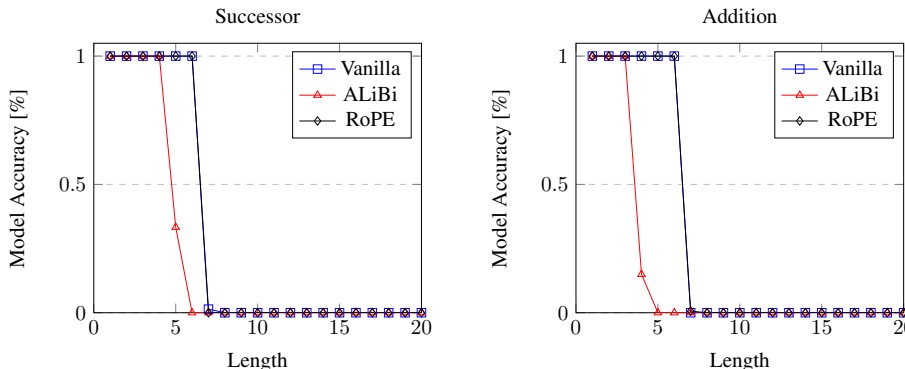

Figure 7: Extrapolation results for models trained on $L_{int} \leq 6$ on **Successor** and **Addition**. Length is measured in the number of digits of one operand.

We observe similar patterns with other tasks. Table 2 summarizes Vanilla Transformer's capabilities for interpolation and extrapolation capabilities on these tasks. We single out the Vanilla model because our ABC scheme works only when the Vanilla model can interpolate.

|  | Interpolation | Extrapolation |
|---|---|---|
| **Successor** | ✓ | ✗ |
| **Addition** | ✓ | ✗ |
| **Parity** | ✗ | ✗ |
| $N \times 1$ | ✓ | ✗ |

Table 2: Vanilla Transformer's interpolation and extrapolation capabilities.

## A.3 ATTENTION ANALYSIS

To figure out the causes of failure to extrapolate, we extract and analyze the attention weights of the vanilla model on **Successor** and **Addition**. Figure 8 gives an example of the attention heat map of one specific head in the last decoder layer during a **Successor** task. Lighter colors represent higher weights.

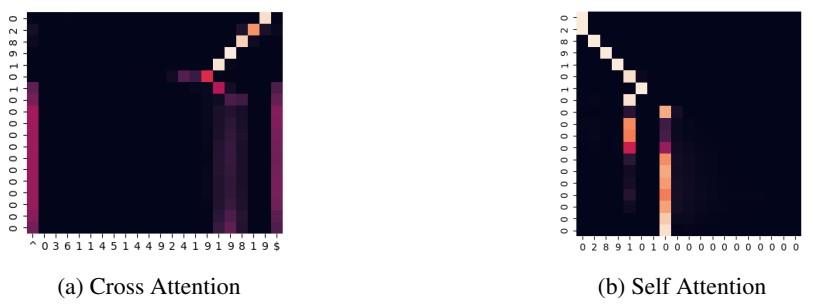

(a) Cross Attention
(b) Self Attention

Figure 8: Attention heat map on "03611451449241919819" for **Successor**.

For the sequence 03611451449241919819, the correct output should be 03611451449241919820. Note that we reverse the output digits during training so the model also generates output starting from the lowest digit and working upwards. The model is correct until the hundred-thousands digit. For an input sequence of length $n$, to generate the $i$-th digit for **Successor** correctly, the crucial information lies in the $(n-i+1)$-th input token and the $(i-1)$-th output token (for possible carry).[7] This means that the correct attention pattern should

---

[7]Note that the tokens are generated in a lowest-digit first order.

light up the "anti-diagonal" (the diagonal from top-right to bottom-left) for the cross attention matrix and "subdiagonal" (the diagonal directly under the main diagonal) for self-attention. From figure 8 it is clear that the Vanilla Transformer correctly learns the attention pattern up to the hundred-thousands digit and fails beyond that. This correlates perfectly with the extrapolation performance shown in figure 7.

For **Addition**, we look at individual heads. Figure 9 shows an example of the attention heat maps of two specific heads in the last decoder layer during an addition task.

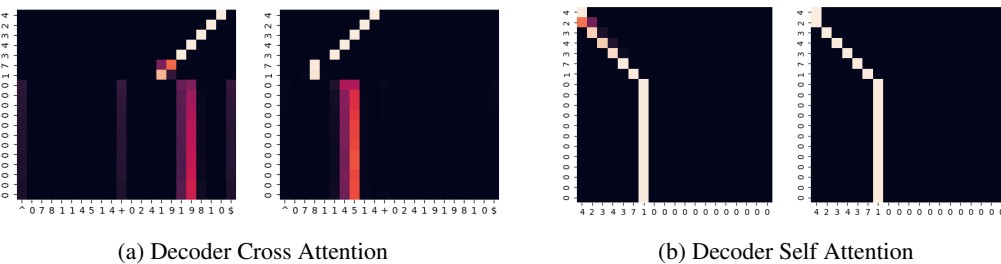

(a) Decoder Cross Attention                    (b) Decoder Self Attention

Figure 9: Attention heat map on "078114514+0241919810" for **Addition**.

In this case we find that there appears to be a sort of differentiation of tasks, where one head looks at the first operand and the other looks at the second. The results are consistent with those found in **Successor**, that the model does a good job identifying which token to attend to up to the maximum training length. Again this echoes with the extrapolation performance of figure 7.

### A.4 ATTENTION BIAS SCAFFOLDING

To future validate our hypothesis, we introduce a number of methods that guide the model to attend to the right places. The ideas are inspired by existing methods for assisting model learning. Those we find effective in arithmetic learning include the following:

**Input Alignment**

When we humans perform arithmetic computations, input alignment is a common practice that facilitates the process. For example, for multi-digit addition, we write the numbers one below the other, aligning them based on place value. We then add from the rightmost digit, propagating through the left, memorizing carries. Without PE/AB, the original Transformer's attention is order-invariant, and, theoretically, the importance of context does not depend on recency. However, certain input representations result in simplified attention patterns that can be captured by the windowed biasing introduced next. Therefore we interleave the digits from the two operands for binary operations so that digits from each operand that should be attended to together are adjacent. Specifically, for a binary operator $\oplus$ (such as +), and two $n$-digit numbers $a = a_n a_{n-1} \ldots a_1$ and $b = b_n b_{n-1} \ldots b_1$ where $a_i$ and $b_i$ are their digits in the proper base representation, the input sequence is transformed as

$$a_n a_{n-1} \ldots a_1 \oplus b_n b_{n-1} \ldots b_1 \longrightarrow \oplus a_n b_n a_{n-1} b_{n-1} \ldots a_1 b_1$$

$N \times 1$ is different since the second operand, say $b$, is single-digit. In this case, we just insert $b$ into the right side of each digit of $a$:

$$a_n a_{n-1} \ldots a_1 \times b \longrightarrow \times a_n b a_{n-1} b \ldots a_1 b$$

Note that input alignment is only used for ABS, to make the attention pattern simply so subsequent methods could "scaffold" the attention to longer inputs more easily. We do *not* need to use it for ABC because ABC could automatically learn the correct patterns. The input to the ABC model is simply the "natural" expression (e.g., 0123+0456 or 0123*6).

**Windowed Attention Biasing**

Biasing towards recency and penalizing attention scores between distant query-key pairs is the basic idea of ABs such as ALiBi (Press et al., 2022). The windowed attention biasing developed by Lonformer (Beltagy et al., 2020) uses a sliding window to control which parts of the attention matrix are "open". We can customize it according to the attention patterns we want to enforce.

Specifically, recall that omitting head indexing, given query, key, and value matrices, the Transformer model (Vaswani et al., 2017) computes attention scores as:

$$Attention(\boldsymbol{Q}, \boldsymbol{K}, \boldsymbol{V}) = \text{softmax}(\frac{\boldsymbol{Q}\boldsymbol{K}^T}{\sqrt{d}})\boldsymbol{V}$$

Let $\boldsymbol{A}_0 = \frac{\boldsymbol{Q}\boldsymbol{K}^T}{\sqrt{d}}$ be the original attention weight matrix before softmax, we bias the weights by $\boldsymbol{A} = \boldsymbol{A}_0 + \boldsymbol{B}_w$, where $w$ is a parameter specifying the window width. The basic idea for constructing $\boldsymbol{B}_w$ is setting a subset of its elements to 0, and the rest to $-\inf$. This essentially masks out certain elements of $\boldsymbol{A}$ to $-\inf$, which, after softmax, results in 0 weights for corresponding tokens,

The construction of $\boldsymbol{B}_w$ depends on the recurrent pattern that encodes the inductive bias about the task (Beltagy et al., 2020). Figure 10 shows the patterns for our tasks. For unary operations, such as successor and parity, generating the current output token depends on the previous output token and one input token at the corresponding position, shown by figure 10 (a). Binary operations, such as addition and $N \times 1$, share the same output token dependency but different input token dependency. In this case, since we align digits from the two operands, as shown in figure 10 (b), the context window spans two consecutive input tokens and also slides two positions at a time.

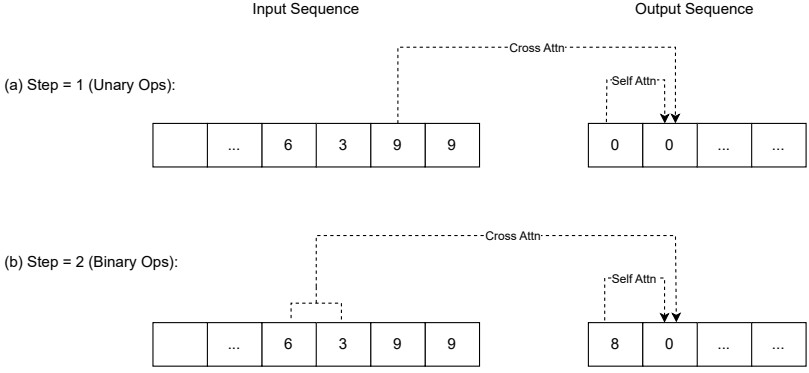

Figure 10: Attention patterns for unary and binary operations.

For an input length $S$ and output length $L$, the bias for decoder self-attention is

$$B_w = \begin{cases} 0, & \text{if } i - k = j \text{ for } i, j = 1, \ldots, L, k = 0, \ldots, w \\ -\inf, & \text{otherwise} \end{cases}$$

That is, the elements of the matrix are all set to $-\inf$ except those on the main diagonal and $w$ elements below. Note that, following the traditional practice (Vaswani et al., 2017) of decoder masking, all elements above the main diagonal are set to $-\inf$ to prevent the decoder from seeing future tokens.

Cross attention bias is similar, with three differences: (1) Since the order of output sequence is reversed, the "open" context windows go along the anti-diagonal direction; (2) Since we align the input digits, the window spans, and also steps over, two positions for binary operations; (3) The open context window extends to both left and right $w$ positions. [8]

Figure 11 is a visualization for the case of $w = 1$.

---

[8] Self attention bias only extends to the left.

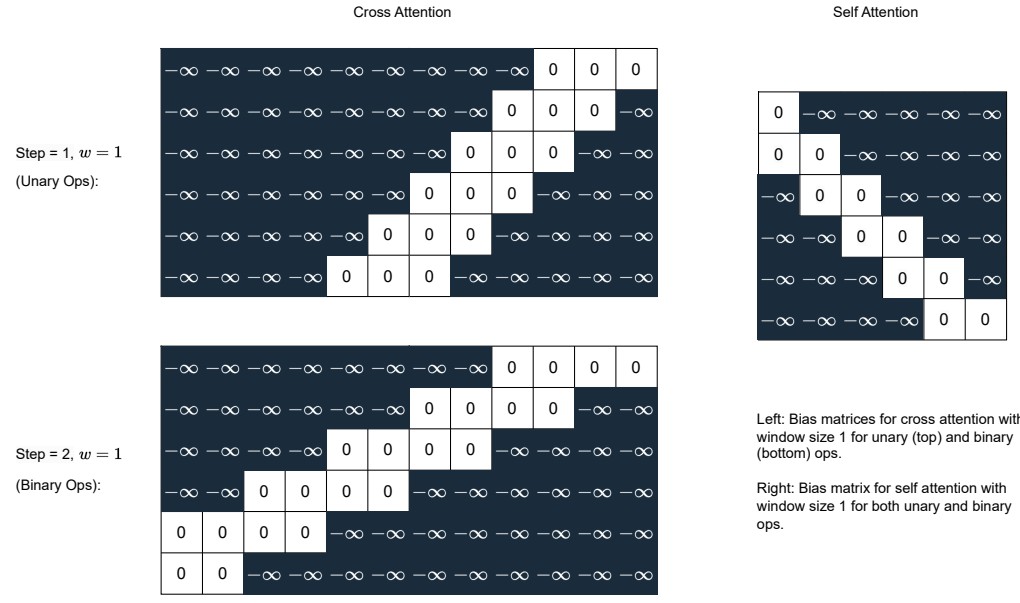

Figure 11: Attention bias matrices for unary and binary operations.

**Cyclic Position Indexing (CPI)**

Position indexing refers to how we identify each individual position. The simplest way is just to index them $0, 1, \dots$. Positional embedding mechanisms are then constructed based on this indexing. Very recently, manipulating position indexing has become an effective and trending method for expanding context windows for Transformer-based LLMs. For example, Chen et al. (2023) and its NTK-aware variant (Peng et al., 2023) modify RoPE with "interpolated" position indices to increase the "density" of positions within the pre-trained context window, thus effectively extending its length.

The motivation for using CPI in our tasks is that large position indices unseen during training may confuse the model. And for arithmetic tasks that admit recurrent generation rules, it is not necessary to identify tokens that are not being currently attended to either. As long as the period is compatible with the context window, it should provide the model with a clear mechanism to differentiate the relevant tokens without diverting its attention. For arithmetic tasks, our empirical study shows that the model is not sensitive to the value of $T$ as long as it produces an open window whose width is approximately that of the bias context window as shown in figure 10. We believe it might be of independent interest for other application secenarios.

A.5    VALIDATION RESULTS

To evaluate the effectiveness of the above mechanisms, we conduct extensive experiments on each of the arithmetic tasks with the following configurations:

(A) **Vanilla**: Vanilla Transformer with sinusoidal PE.

(B) **+ $w = 1$**: (A) + windowed attention biasing with $w = 1$.

(C) **+ $T = 3$**: (B) + additional CPI with a period of $T = 3$.

(D) **NoPE + $w = 1$**: Windowed attention biasing only, without PE at all, with $w = 1$.

We experimented with a few different $w$ and $T$ values and found that slight variations do not produce very different results thus we report the best-performing configurations above.

Results are presented in table 3. None of the previous works achieves extrapolation on any of the tasks. RPE (Jelassi et al., 2023) maintains 90+% accuracy up to 20 digits but does not go beyond. Vanilla Transformer and RoPE could interpolate, achieving 100% accuracy for 6-digit inputs, for all

Table 3: Extrapolation results measured as percent accuracy (%). Numbers in bold show the best accuracies achieved for the corresponding input length limit.

| Task | Model | Length (Number of Digits) | | | | |
|---|---|---|---|---|---|---|
| | | 6 | 10 | 15 | 20 | 50 |
| Successor | Vanilla | **100.0** | 0.0 | 0.0 | 0.0 | 0.0 |
| | $+ w = 1$ | **100.0** | **100.0** | **100.0** | **100.0** | **100.0** |
| | $+ T = 3$ | **100.0** | **100.0** | **100.0** | **100.0** | **100.0** |
| | NoPE $+ w = 1$ | **100.0** | **100.0** | **100.0** | **100.0** | **100.0** |
| | ALiBi | 1.3 | 0.0 | 0.0 | 0.0 | 0.0 |
| | RoPE | **100.0** | 0.0 | 0.0 | 0.0 | 0.0 |
| Addition | Vanilla | **100.0** | 0.0 | 0.0 | 0.0 | 0.0 |
| | $+ w = 1$ | **100.0** | 0.0 | 0.0 | 0.0 | 0.0 |
| | $+ T = 3$ | **100.0** | **100.0** | **100.0** | **100.0** | **100.0** |
| | NoPE $+ w = 1$ | 99.95 | 99.81 | 99.84 | 99.76 | 99.35 |
| | ALiBi | 0.0 | 0.0 | 0.0 | 0.0 | 0.0 |
| | RoPE | **100.0** | 0 | 0 | 0 | 0 |
| | RPE* | **100.0** | 99.9 | 97.2 | 21.3 | N/A |
| Parity | Transformer† | | 52.00†/52.60‡ | | | |
| | $+ w = 1$ | **100.0** | **100.0** | **100.0** | **100.0** | **100.0** |
| | $+ T = 3$ | **100.0** | **100.0** | **100.0** | **100.0** | **100.0** |
| | NoPE $+ w = 1$ | **100.0** | **100.0** | **100.0** | **100.0** | **100.0** |
| $N \times 1$ | Vanilla | **100.0** | 0.0 | 0.0 | 0.0 | 0.0 |
| | $+ w = 1$ | **100.0** | 6.0 | 0.19 | 0.0 | 0.0 |
| | $+ T = 3$ | **100.0** | **100.0** | **100.0** | **100.0** | **100.0** |
| | NoPE $+ w = 1$ | 99.89 | 99.63 | 99.49 | 99.39 | 98.31 |
| | RoPE | **100.0** | 0 | 0 | 0 | 0 |

\* Data taken from Jelassi et al. (2023) which is an encoder-only architecture with shared layers.
† Data taken from Deletang et al. (2023) which evaluates five encodings (none, sin/cos, RoPE, ALiBi, and the relative positional encoding from Transformer-XL) and reports the best-performing variant.
‡ Data taken from Ruoss et al. (2023) which uses randomized positional encodings to boost length generalization.

the tasks. ALiBi does not even interpolate. Its accuracies drop to near 0s on all tasks beyond 3 or 4 digits (figure 7).

On the other hand, our solutions (windowed attention biasing + CPI) achieve complete length generalization on all tasks, maintaining 100% accuracy up to 50 digits. Unary tasks (**Successor** and **Parity**) appear to be not relying on any positional embedding at all once the windowed attention biasing is in place, which is also robust against possible perturbation of any PE.

For binary tasks (**Addition** and $N \times 1$), on the other hand, there appears to be some bad interaction between the original sinusoidal PE and windowed attention biasing. Both the original sinusoidal PE and $+ w = 1$ (sinusoidal PE with windowed bias) configurations only achieve interpolation but not extrapolation. Windowed biasing without any PE at all (NoPE+$w = 1$) results in a slightly imperfect generalization for both binary tasks.

## A.6   THE CASE OF PARITY

The case of **Parity** deserves more explanation. **Parity** is a well-known failure mode for Transformers. Hahn (2020) shows that, for self-attention models with soft attention (e.g., a typical Transformer), the change in activation at the decoder layer that changing a single input symbol can cause is bounded by $O(1/n)$ where $n$ is the input length (Lemma 5). This means that for tasks that are

sensitive to changes of a small number of input symbols, such as **Parity**, Transformer models cannot make accurate predictions for long sequences, as softmax is unable to produce very different predictions for inputs that result in similar activations. This is consistent with other works such as Deletang et al. (2023), all of which obtain performances slightly better than random guessing. Deletang et al. (2023) prove that there exists a Transformer construction that can achieve perfect Parity, but such a construction is not learnable.

Ours is the *first* learnable Transformer that obtains perfect accuracy and length generalization for the **Parity** task. We overcome the difficulties of Hahn (2020) by applying the "scaffolding" methods introduced earlier. Mathematically, our attention guiding can be seen as allowing the model to produce (partial) results based on only local tokens, effectively reducing $n$ to a constant window size.

# B    ALGORITHM FOR ABC

---

**Algorithm 1** Attention Bias Calibration (ABC) for non-negative $\Delta$ [9]

---

**Input**:

$\boldsymbol{A}_{in}$: The attention tensor with dimensions $[H, m, n]$, where $h$ represents the number of heads and $m$, $n$ represents the number of rows and columns in each attention matrix, respectively.

$M, N$: The dimensions of the output bias matrix

$\mathbb{D}$: A set of tuples $(1, \Delta)$. It represents the set of all directions we want to search for patterns.

**Output**: $\tilde{\boldsymbol{A}}$, a tensor with the dimensions $[H, M, N]$, representing the bias matrix for each head.

1: **for** $h \leftarrow 1$ to $H$ **do**
2:    **for** $(1, \Delta) \in \mathbb{D}$ **do**                                    ▷ Iterate Directions
3:       **for** $i \leftarrow 1$ to $M$ **do**
4:          **for** $j \leftarrow 1$ to $N$ **do**
5:             **while** $k + i \leq m$ **and** $k\Delta + j \leq n, \ k \in \mathbb{Z}$ **do**
6:                $\tilde{\boldsymbol{A}}_{tmp}[h][(1, \Delta)][i][j] + = \boldsymbol{A}_{in}[h][k + i][k\Delta + j]$
7:                $size + = 1$
8:             **end while**
9:             $\tilde{\boldsymbol{A}}_{tmp}[h][(1, \Delta)][i][j]/ = size$          ▷ Average diagonals (if $size \neq 0$)
10:          **end for**
11:       **end for**
12:       **for** $i \leftarrow 1$ to $M$ **do**
13:          **for** $j \leftarrow 1$ to $N$ **do**
14:             $\tilde{\boldsymbol{A}}_{tmp}[h][(1, \Delta)][i][j] \leftarrow \tilde{\boldsymbol{A}}[h][i][j] - \max(\tilde{\boldsymbol{A}})$          ▷ Normalize
15:          **end for**
16:       **end for**
17:       **for** $i \leftarrow 1$ to $M$ **do**
18:          **for** $j \leftarrow 1$ to $N$ **do**
19:             $\tilde{\boldsymbol{A}}_{tmp}[h][(1, \Delta)][i][j] \leftarrow dropout(\tilde{\boldsymbol{A}}[h][i][j])$          ▷ Dropout
20:          **end for**
21:       **end for**
22:    **end for**
23:    **for** $i \leftarrow 1$ to $M$ **do**
24:       **for** $j \leftarrow 1$ to $N$ **do**
25:          $\tilde{\boldsymbol{A}}[h][i][j] \leftarrow \max(\tilde{\boldsymbol{A}}_{tmp}[h][(1, \Delta)][i][j], \tilde{\boldsymbol{A}}[h][i][j)$          ▷ Merge directions
26:       **end for**
27:    **end for**
28: **end for**
29: **return** $\tilde{\boldsymbol{A}}$

---

## C   SOLUTION TO SINGLE-OPERATOR, SINGLE-DEPTH LISTOPS TASK

A single-operator, single-depth ListOps task is a ListOps task restricted to a single operator and depth 1. The problem essentially reduces to a prefix arithmetic task of the form

$$\oplus x_n x_{n-1} \ldots x_1 \to y$$

where $\oplus$ is the operator, and $x_i \in \{0, 1, \ldots, 9\}$. In this task, we only support operators that are associative. They include SM, MAX, and MIN.

Since the operators are all associative, we turn this task into a sequence-to-sequence form using the idea similar to how we handle the parity problem. That is, we start with the right-most argument, and work to the left digit by digit, generating intermediate results and appending them to the target:

$$y_1 = x_1, y_i = y_{i-1} \oplus x_i, i = 2, \ldots, n.$$

This is in the exact same form as the Parity problem, except in decimal instead of binary, and we solve it using the same windowed attention with $w = 1$ and $T = 3$.

---

[9]The algorithm for negative $\Delta$ is identical except that, before invoking the same procedure, we translate $\boldsymbol{A}_{in}$ $N - n + 1$ elements to the right so that the top-right corners of $\boldsymbol{A}_{in}$ and $\tilde{\boldsymbol{A}}$ align.

