# OpenReview forum: "From Interpolation to Extrapolation: Complete Length Generalization for Arithmetic Transformers"
_ICLR.cc/2024/Conference — Submitted to ICLR 2024_

### Official Review · Reviewer_hH7Y · 2023-10-27

**Soundness:** 3 good
**Presentation:** 3 good
**Contribution:** 3 good
**Rating:** 6
**Confidence:** 4

**Summary:**

The authors achieve perfect length generalization on algorithmic tasks (x+1, x+y, and N x 1). The models are trained with as little as a sequence length of 6, and they generalize to sequences with a length of 5. They achieve this by learning an attention pattern on short sequences which interpolates and extrapolating the attention pattern to long sequences, in the form of an additive bias. This bias is somewhat learned, however it has a very strong structural prior in the form of lines in the attention matrix.

**Strengths:**

- Perfect length generalization in addition
- Clear presentation
- The method is based on the analysis of the attention patterns
- The paper has a clear scope
- The method clearly shows that the attention pattern is the bottleneck in extrapolating to longer sequences

**Weaknesses:**

- The inductive bias is very strong: it biases the attention matrix to be diagonal lines. The method might not work in more general cases in 2 different ways: it might not help with generalization, but it might also hinder learning (I strongly suspect that this would be the case with language-related tasks). Thus, the limitations of the method are not clear. I think this is the biggest weakness of the paper.
- The authors already compare to lot of baselines. However, these relative positional encodings are known to have extrapolation issues. The authors also cite references for this. However, the authors do not cite some relevant work that has been shown to help with extrapolating to longer sequences [1, 2], for example by Transformer-XL style attention [3]. Other work already showed extrapolation on some algorithmic tasks, for example, ListOps and modular arithmetic [4]. Please cite it.

1. Csordas et al, 2022: The Devil is in the Detail: Simple Tricks Improve Systematic Generalization of Transformers
2. Ruoss et al, 2023: Randomized Positional Encodings Boost Length Generalization of Transformers
3. Dai et al, 2019: Transformer-XL: Attentive Language Models Beyond a Fixed-Length Context
4. Csordas et al, 2021: The Neural Data Router: Adaptive Control Flow in Transformers Improves Systematic Generalization

**Questions:**

- Adding a task where the attention pattern should not be a straight line would be interesting, to see the limitations of the method. One interesting task could be ListOps (for an analysis of attention patterns, see [4] from the "Weaknesses" chapter)
- On page 6, when explaining how the attention matrices are extracted, it is written "are parameter matrices in the last decoder layer". Does this mean that only the last decoder layer has the attention bias? Or do all of them have it, but set to the same as the last layer?
- The absolute PEs are usually not injected in the attention head directly, but to the residual stream immediately after the token embeddings. Given this, in the first eq. on P6, what are the additive p_i and p_j components? Do you do a custom method, where you inject the PE directly into the attention at each layer?
- How do you handle the cross-attention? Does that have this bias as well?
- Have you tried a less aggressive bias, where you don't set the elements to -inf, but to some finite negative number, such that the network has the chance to overcome the bias if it is good for the task? Do you think it would work in this case?

I'm willing to raise my score if my questions about the positioning of the biases are cleared, and if the authors could run their method on any task that does not require strictly monotonic attention (e.g ListOps), regardless of the outcome of the experiment. Just to make it clear what are the limitations.

---

> ### Author Response · Authors · 2023-11-22
>
> Thank you for your insightful review and valuable feedback. We have revised the paper based on your suggestions. Before we address your specific questions, here is a summary of the improvements we made to the paper:
>
> 1. We simplified the extensive notations used in section 5 and added more visuals to explain the concept more clearly.
> 2. In section 8, we added a discussion on the scope and limitations of our work to clarify its applicability.
> 3. We conducted additional experiments with ListOps and discussed the results in section 8 and appendix C.
> 4. We added citations to the works in Systematic Generalization. Thanks for pointing them out!
> 5. We improved section 4 by showing the attention patterns in the main text.
>
> **Adding a task where the attention pattern should not be a straight line would be interesting**
>
> Thank you for such a great suggestion. ListOps is an excellent test case for learning more complex structures. We conducted additional experiments with ListOps and discussed the results in section 8 and appendix C. Unfortunately, with the model scales we experimented with and the amount of resources we had, we could only solve a simplified single-operator, single-depth ListOps task via ABS. Learning multi-depth ListOps may require scaling up the model dramatically, as the heuristic in the NDR paper [2] indicates. Our goal is certainly to study the learning of more complex tasks and we plan them as future research.
>
> **Does this mean that only the last decoder layer has the attention bias? Or do all of them have it, but set to the same as the last layer?**
>
> We apologize for the confusion. For the tasks presented in this work, the bias matrices are obtained from the last decoder layer and applied to all layers during re-train, on a per-head basis. More flexible configurations such as per-layer bias could work better for more complex tasks, which is what we currently are exploring. We added clarification on this issue at the end of section 5.
>
> **Given this, in the first eq. on P6, what are the additive p_i and p_j components? Do you do a custom method, where you inject the PE directly into the attention at each layer?**
>
> Again we apologize for the confusion.  We apply PE at the start of the process as the original Transformer paper does [1], rather than injecting it at each layer. The notation we used is misleading and we have revised it in section 5. Thanks for pointing it out.
>
> **How do you handle the cross-attention? Does that have this bias as well?**
>
> We used ABC on cross-attention as well. It works the same way as self-attention, and Figures 4 and 5 are both cross-attention patterns.
>
> **Have you tried a less aggressive bias, where you don't set the elements to -inf, but to some finite negative number, such that the network has the chance to overcome the bias if it is good for the task? Do you think it would work in this case?**
>
> This is an excellent idea and we are in the process of trying it out on NLP tasks which we think is a more suitable situation. We believe that in NLP, ABC could be possibly used to capture some global or background dependency patterns. However, the patterns are likely to be less rigid and clear than those in arithmetic tasks. A "soft" ABC, i.e., something like
> $$attn = (1-\lambda) \cdot attn_0+\lambda \cdot bias$$
> where bias acts as a global context while $attn_0$ captures the local dependency, might work better. This is our ongoing work.  Thanks for your suggestion!
>
> **If the authors could run their method on any task that does not require strictly monotonic attention**
>
> We have run experiments on ListOps which we discussed earlier.
>
> Thanks again for the constructive feedback that helped make this paper better! We hope our responses addressed your concerns.
>
> **References:**
>
> [1] Attention is All You Need
>
> [2] The Neural Data Router: Adaptive Control Flow in Transformers Improves Systematic Generalization

---

> ### Comment · Reviewer_hH7Y · 2023-11-22
>
> I would like to thank the authors for their updates and clarifications. Most of my questions have been answered.
>
> I find the outcome of the ListOps experiments very surprising. The authors use a simplified version of ListOps, where there is only a single operation applied to a list of numbers. My original intent was the opposite: to run it on recursively embedded operations, to see if the model can allow non-monotonic attention patterns. The surprising part of the current results is that for this simplified version of ListOps, monotonic memory access patterns should be enough, so I think the method should work (and my prediction was that it would fail on the original, recursive version). Given these negative outcomes, I think the authors should emphasize the limitations of the paper more so that it can be clear to the reader that this is not a ready-to-go solution for length generalization.
>
> The other remaining weakness is not compared with relative positional encodings.
>
> Overall, I think the paper contributes an interesting new idea. I consider it worth publishing even if it is not clear yet if it is widely applicable to other tasks. It could serve as a base for interesting future research.

---

### Official Review · Reviewer_25Z2 · 2023-10-30

**Soundness:** 3 good
**Presentation:** 2 fair
**Contribution:** 3 good
**Rating:** 6
**Confidence:** 4

**Summary:**

The authors propose an analysis of length generalization in sequence to sequence transformers for four arithmetic tasks: successor (adding 1), parity integration, addition, and multiplication by one digit operands.

In a first set of experiments, they observe that on these four operations, length generalization can be achieved if the attention window for the model is constrained to focus on a small number of tokens, either directly, by forcing the window to be of width one, or indirectly, by using small modular integers for positional encoding. On these tasks, such constrained attention models can effectively replace positional encodings.

Finally, the authors propose a technique for creating such attention constraints for a specific task. To this effect, they average the biases in attention weights when a model, trained to generalize to sequence of the same length, performs the task. Then, they retrain on the specific task, using these biases to constrain attention. Experiment indicate that this allows the model to generalize to longer sequences for three of the four tasks.

**Strengths:**

The paper focuses on an important limitation of transformers: length generalization. The analysis of the role of the attention window is compelling, and the experimental results from appendix A.5 are quite convincing.

The calibration technique proposed in section 5 is interesting, and experiments demonstrate its worth on the three tasks considered.

Overall, the paper provides is a solid analysis of length generalization, and interesting ideas on how to solve it.

**Weaknesses:**

* The important results from section 4 are hard to understand if one only reads the main paper. In particular, the claim that constrained attention solves parity is only supported in the appendix. Also the logic behind section 5 is hard to comprehend without reading parts of section A.4 (the precise description of windowed attention, and the cyclic encoding). The paper would be made much stronger by moving a significant part of appendices A.4 and A.5 into section 4. This can probably be done at the expense of figure 1 (not very informative), and parts of section 5.

* The claim that constrained attention, and ABC, offers a **complete** solution to length generalization on problems of arithmetic is exaggerated. All the tasks considered in the paper share a common trait: their solution can be computed by looking at just a few consecutive digits of the problem and the currently computed solution. This would break for such a basic arithmetic task as summing $n$ k-digit integers, and generalizing on larger $n$. As for multiplication, $N\times 1$ multiplication is closer to $10$ unary operation than a binary operation. $N\times N$ multiplication would have the same non local behavior.

* Given the comments in section 7, about the similarity between ABC and RPE, it is regrettable that the experiments in section 4 do not feature RPE (except addition, which corresponds to a different architecture). It would be interesting, in particular, to see how RPE perform on the successor and parity task.

**Questions:**

* You use 1/6 layer transformers, what is the benefit of a shallow encoder, compared to a (smaller) 6/1 encoder, or 4/4 model?
* All your vanilla models use the cosine positional embeddings. You observe that it sometimes interferes with the sliding window attention, could you try learnable PE?
* Could you try RPE on the successor and parity experiments? Your conclusions suggest that it would help generalize.
* In the experiments from appendix A.5 (table 3), you try cyclic PE on top of a w=1 windowed attention, what happens if you use it without windowed attention? (it is another form of RPE)
* Have you tried $N\times N$ multiplication (for smaller values of $N$)? of $N\times k$ for $k$ larger than 3 (so that the model cannot memorize all cases as so many unary operations)?
* The Jelassi results in table 3 and figure 1 correspond to a different architecture (encoder-only, with shared layers), it would be better to rerun them, or at least to clarify this.

---

> ### Author Response · Authors · 2023-11-22
>
> Thank you for your insightful review and valuable feedback. We have revised the paper based on your suggestions. Before we address your specific questions, the following is a summary of the improvements we made to the paper:
>
> 1. We simplified the extensive notations used in section 5 and added more visuals to explain the concept more clearly.
>
> 2. In section 8, we added a discussion on the scope and limitations of our work to clarify its applicability.
>
> 3. We conducted additional experiments with ListOps and discussed the results in Section 8 and Appendix C.
>
> 4. We added citations to the works in Systematic Generalization.
>
> 5. We improved section 4 by showing the attention patterns in the main text.
>
> **The important results from section 4 are hard to understand if one only reads the main paper.**
>
> We agree that section 4 was a bit crammed and some material was left in Appendix A. We have reformatted sections 4 and 5 to be more clear and easy to understand.
>
> **The claim that constrained attention, and ABC, offers a complete ...**
>
> We apologize for the confusion. By "complete length generalization" we didn't mean that we solved all length generalization problems. Instead, we meant that our solution solves some specific arithmetic algorithm learning tasks, allowing the model to generalize (completely) to long lengths on those tasks. Ours was the first solution to achieve such results. We clarified the notion in the paper and added a discussion on the scope and limitations of our work in section 8.
>
> **You use 1/6 layer transformers, what is the benefit of a shallow encoder, compared to a (smaller) 6/1 encoder, or 4/4 model?**
>
> We have conducted tests with 6/6, 3/6, 3/3, and 1/3 transformers. We see little to no difference regarding both interpolation and extrapolation performances. We suspect that this is due to the simplicity of our tasks.
>
> **You observe that it sometimes interferes with the sliding window attention, could you try learnable PE?**
>
> Your suggestion to examine learnable positional encodings (PE) is well-taken, but we did not try learnable PE because studies have shown that learnable PE does not achieve good results when doing length extrapolation. More specifically, as [1]  mentioned, "We also experimented with using learned positional embeddings instead, and found that the two versions produced nearly identical results". The learnable PE referenced here cannot enable extrapolation because all the encodings are learned on the in-training length. Our decision not to include learnable PEs in our experiments was also influenced by hardware limitations.
>
> **Could you try RPE on the successor and parity experiments? Your conclusions suggest that it would help generalize.**
>
> The potential of relative positional encodings (RPE) to improve length generalization is a promising direction for future research. While we have not included RPE in our current experiments, we have contrasted ABS with the top-performing variant of RPE as per [2] in the appendix, showing the superiority of ABS in Parity.
>
> **In the experiments from Appendix A.5 (table 3), you try cyclic PE on top of a w=1 windowed attention, what happens if you use it without windowed attention?**
>
> We also think that cyclic PE has some potential to act as a sort of RPE. However, when we tested cyclic position indexing (CPI) without attention windows, it did not yield satisfactory results and the model did not even interpolate. CPI appears effective only on very short windows.
>
> **Have you tried $n×n$ multiplication (for smaller values of $n$)? of $n×k$ for $k$ larger than 3 (so that the model cannot memorize all cases as so many unary operations)?**
>
> We have attempted $n×n$ multiplication and $n×k$ multiplication with $k$ larger than 3. Unfortunately, these experiments did not yield successful interpolation, rendering ABC unusable in this context. We have addressed this limitation in the revised paper in section 8.
>
> **The Jelassi results in Table 3 and Figure 1 correspond to a different architecture (encoder-only, with shared layers), it would be better to rerun them, or at least to clarify this.**
>
> We apologize for the misleading caption, this has been revised.
>
> Thank you for your insightful response.
>
> **References:**
>
> [1] Attention is All You Need
>
> [2] Neural Networks and the Chomsky Hierarchy

---

> > ### Comment · Reviewer_25Z2 · 2023-11-22
> >
> > I thank the authors for their thoughtful responses. I will take them into account in the upcoming discussion with other reviewers and area chairs.

---

### Official Review · Reviewer_usKR · 2023-10-31

**Soundness:** 3 good
**Presentation:** 2 fair
**Contribution:** 2 fair
**Rating:** 5
**Confidence:** 4

**Summary:**

This paper studies length generalization of Transformer models on algorithmic tasks. It shows that the vanilla Transformer correctly learns the attention patterns up to the training length and fails beyond that. To address this problem, the authors propose Attention Bias Calibration (ABC), which introduce an additional stage to obtain proper attention biases from training data. Experiments show the effectiveness of the proposed approach.

**Strengths:**

- The experiment results are good on the tasks considered in this paper.
- The authors provide some interesting empirical study and visualization of the attention.

**Weaknesses:**

- The presentation of this paper looks good in the first two sections, but the presentation quality drops in the subsequent parts. In particular, section 5 is very hard to read due to the intensive definition and sometimes abuse of notations albeit the idea is very simple. The definitions of some notations are far from where it is used. For example, the authors should have mentioned how $threshold$ is calculated when it appears for the first time – In the paper, it is not defined until section 6, which can confuse the readers.
- The method implicitly encodes sparse attention patterns, since the attention bias $\tilde A_{i,j}$ can be set to $-\infty$ for a large number of tokens. While it can be helpful for simple tasks, I’m not convinced of its effectiveness in more general settings (e.g., for natural language). I'm worried ABC can only be effective for a small number of tasks. The tasks in this paper also seemed to be oversimplified. I'm wondering if ABC can still work well for multi-digit multiplication rather than the simple $N\times 1$.
- ABC is only compared against weak baselines in the experiments. As discussed in the paper, existing papers have shown Alibi and RoPE are suboptimal on algorithmic tasks. The authors should compare ABC against stronger baselines, e.g., Randomized Positional Encodings, RoPE with Position Interpolation, etc.
- Typos.
  - Footnote 4, “16 etc” $\to$ “16, etc”.
  - Footnote 6, “tasks” $\to$ “tasks.”.

**Questions:**

- Can ABC still work well for more complex task, e.g., multi-digit multiplication, or even natural language modeling?

- Can you compare ABC with more advanced baselines?

---

> ### Author Response · Authors · 2023-11-22
>
> Thank you for your insightful review and valuable feedback. We have revised the paper and the following is a summary of the main changes:
>
> 1. We simplified the extensive notations used in section 5 and added more visuals to explain the concept more clearly.
> 2. In section 8, we added a discussion on the scope and limitations of our work to clarify its applicability.
> 3. We conducted additional experiments with ListOps and discussed the results in Section 8 and Appendix C.
> 4. We added citations to the works in Systematic Generalization.
> 5. We improved section 4 by showing the attention patterns in the main text.
>
> Now addressing your questions:
>
> **The presentation of this paper looks good in the first two sections, but the presentation quality drops in the subsequent parts.**
>
> We have revised Section 5, reducing the complexity of notations and augmenting the textual explanations with additional visuals to facilitate a clearer understanding of the concepts presented.
>
> **While it can be helpful for simple tasks, I’m not convinced of its effectiveness in more general settings (e.g., for natural language)**
>
> We added a discussion on the scope and limitations of our work in section 8. The current sparse attention bias works for the arithmetic tasks we study. For more complex tasks, it is likely to require a scaling up of the model, as the heuristic in the NDR paper [3] indicates, and much more expressible ways to implement the bias. Although exploring ABC in more generalized settings is beyond the scope of this current paper, we intend to address this in our subsequent research. Sparse global attention has been shown to have a great effect on the transformer model's performance. More specifically, in [1], they demonstrated that the transformer model can still hold their abilities even when the entire attention mechanism is replaced with a static attention matrix generated from averages of the attention matrix across a corpus of data, only suffering from an 8% decrease in accuracy in some cases. These findings, alongside ABC’s conceptual similarity with Relative Positional Encodings (RPE), provide a promising foundation for future investigations into ABC's utility in complex NLP tasks. We are currently in the process of exploring more tasks in NLP.
>
> **Can you compare ABC with more advanced baselines?**
>
> The suggestion to benchmark ABC against more sophisticated baselines is well-taken. While ABC has shown remarkable performance in the experiments conducted, outperforming all relevant baselines in [2] and [4] and achieving exceptional accuracies of 99+ percent over extended sequence lengths (over 8 times the training length), we acknowledge the value of broader comparisons. The benchmarks that we compare against include results from [2] and [4], which are very recent works and include extensive experiments. For example, [4] considered five major positional encodings: none, classical sin/cos, RoPE, ALiBi, and the relative positional encodings of Transformer-XL, and report the best-performing configurations.  We believe they are strong SOTA results. Yet the transformer did poorly on those tasks. This indicates that the current state of transformers on arithmetic learning is rather weak. And ABC enables the model to generalize on certain tasks which we believe is a big step forward.
>
> **References:**
>
> [1] How Much Does Attention Actually Attend? Questioning the Importance of Attention in Pretrained Transformers
>
> [2] Randomized Positional Encodings Boost Length Generalization of Transformers
>
> [3] The Neural Data Router: Adaptive ControlFlow in Transformers Improves Systematic Generalization
>
> [4] Neural Networks and the Chomsky Hierarchy

---

> > ### Comment · Reviewer_usKR · 2023-11-23
> >
> > I appreciate the authors' responses and clarifications. However, several concerns remain:
> >
> > - The major concern is that ABC only solves very simple tasks that are carefully chosen. It even fail on moderately complex tasks like ListOps. Without experiments on larger models, it remain doubtful to me whether the failure is because of model size or fundamental limitations of ABC. Given current scope of the experiments, I don't think ABC is a big step forward in arithmetic Transformers.
> >
> > - On the baseline choices, I understand that the authors use the positional encoding methods in [1] as baselines. However, all these methods appeared on or before 2021. I reiterate that the authors should also compare ABC with more recent methods like Randomized Positional Encodings [1], RoPE with Position Interpolation [2], etc.
> >
> > [1] Randomized Positional Encodings Boost Length Generalization of Transformers
> >
> > [2] Extending Context Window of Large Language Models via Positional Interpolation
> >
> > Given these reasons, I maintain my original rating.

---

### Official Review · Reviewer_69hA · 2023-10-31

**Soundness:** 2 fair
**Presentation:** 2 fair
**Contribution:** 2 fair
**Rating:** 5
**Confidence:** 4

**Summary:**

The paper explores how transformer models can learn arithmetic algorithms and achieve optimal length generalization. It introduces attention bias calibration (ABC) to guide the model to focus on the right tokens. Using ABC, the Transformer model can perfectly generalize on certain arithmetic tasks. Finally, the paper also makes a connection between ABC and relative position encoding (RPE).

**Strengths:**

The raised research problem is well-motivated and interesting. The authors show that additional training time is negligible in the paper.

**Weaknesses:**

1. The writing should be improved in terms of clarity. Several terms and concepts are introduced without adequate definitions or elaboration. Specifically, the terms "complete length generalization" and "the organic Transformer" seem ambiguous.

2. ABC constructs attention biases based on task-specific data. It's not clear if it can be applied in multi-task learning settings or even serve as a building block for general-purpose language models.

3. More experiments should be conducted. While the results are promising on the four tasks detailed in Sec. 3.1, more extensive experiments are anticipated. It might be worthwhile for the authors to explore tasks in [1,2].

4. Factual errors in the discussion on related works. Sec. 2 mentions that [3] is a follow-up of Alibi. This can’t be true because [3] appears much earlier than Alibi. Sec. 7 mentions that “Interestingly, such clipping is also used in RoPE”. I believe this claim is incorrect and can be misleading.

[1] Neural networks and the chomsky hierarchy.

[2] Randomized Positional Encodings Boost Length Generalization of Transformers.

[3] Longformer: The long-document transformer.

**Questions:**

Please see weakness above.

---

> ### Author Response · Authors · 2023-11-21
>
> Thank you for your valuable feedback! We have uploaded a revised version of our paper and will address your questions below:
>
> **Writing Clarity:**
> We have made changes to improve readability and make the line of reasoning more clear. In particular, we simplified the extensive notations used in section 5 and added more visuals to explain the concept more clearly.
>
> **ABC's Generality and Use Cases:**
> In section 8, we added discussion on the scope and limitations of our work to clarify its applicability. Ours is an initial attempt to study the roles of attention in transformer’s learning. To this end, we need to find tasks on which transformers can generalize. Surprisingly, even for tasks as simple as addition, no previous transformer model ever succeeded. We take this as an indication that our understanding of the model’s learning mechanism is inadequate. Therefore, the successful cases obtained in this paper, either via ABS or ABC, even though only on simple recurrent patterns and task-specific models, solve a few long-standing difficult or even “impossible” tasks (e.g., Parity) and represent a significant step forward.
>
> **Additional Tasks:**
> Additional and more complex tasks appear to require composition of simple patterns. Learning such compositions may require scaling up the model dramatically, as the heuristic in the NDR paper (Csordás et al., 2021b) indicates. We are exploring them in our current research.
>
> **Factual Errors:**
> You are right on both points, and we have made appropriate modifications in the revision. Thank you very much for pointing them out!  Those are very keen observations!
>
> Again, we thank you for your valuable feedback and we hope that our answers and revision addressed your questions.

---

### Meta-Review · Area_Chair_kVWh · 2023-12-06

**Metareview:**

This work proposes a certain modification to standard Transformer architecture and training, which they show can significantly improve length generalization for certain arithmetic tasks (e.g. addition). The modification essentially involves looking at certain aggregate-statistics of the attention maps on shorter-length inputs, and extending these to attention maps of longer inputs. This proposal is found to improve length-generalization on certain simple tasks (e.g. addition and successor), but not for others (multi-digit multiplication and parity).

Reviewers agreed that the proposed method is interesting, and that the work focuses on an important limitation of Transformers.
However, the reviewers observed several significant weaknesses which were not fully addressed by the rebuttals.
Weaknesses:
* The proposal enforces a very strong inductive bias on the Transformer model. Strong inductive biases are generally expected to help on certain tasks, at the expense of other tasks — understanding the tradeoff is therefore critical. However, this work does not conduct a thorough study of which tasks are affected (positively or negatively) by the attention biasing. This limits its significance, because it is unclear whether the modification will say, improve addition, at the expense of many other tasks. It is also unclear how this proposal affects standard natural-language training.
* The experiments only show improved generalization on tasks with a very simple structure. As Reviewer 25Z2 notes, these are tasks with constant query-complexity of the next-token function.
* The framing is over-claimed: it claimed to yield “unprecedented perfect length generalization on certain arithmetic tasks.” Even the results in Table 1 are not technically “perfect”, and improvements to length-generalization are not “unprecedented.” (As noted by several reviewers, there are many prior works which improve length-generalization by some amount).
* There are various technical shortcomings with the experimental results. Table 1 does not consider RPE for successor & multiplication, and the RPE results reference an experiment from prior work with a different architecture. It is unclear whether or not such missing details matter without presenting controlled experiments. This is especially relevant because the cited paper of Jelassi et al. (2023) also finds some type of length-generalization under a related setup.

These weaknesses outweigh the strengths of this paper, so I must recommend rejection. Since the idea is promising, I encourage the authors to continue their investigation, and take into account all reviewer feedback for future submissions.

**Justification For Why Not Higher Score:**

The weaknesses above are significant enough to prevent acceptance.

**Justification For Why Not Lower Score:**

N/A

---

### Decision · Program_Chairs · 2024-01-16

Reject